# The Validation of the Persian Version of the ID-Migraine Questionnaire

**DOI:** 10.3390/medsci13040213

**Published:** 2025-10-01

**Authors:** Amirreza Nasirzadeh, Amir Ghasemi, Alireza Afshari-Safavi, Mohammad Ali Nahayati, Reza Jahanshahi, Erfan Yavari, Mahan Farzan, Mahour Farzan, Asghar Bayati, Faraidoon Haghdoost

**Affiliations:** 1Department of Medical-Surgical Nursing, Faculty of Nursing, Nursing Research Center, Gonabad University of Medical Sciences, Gonabad 9691880002, Iran; nasirzadeharnz@gmail.com; 2School of Nursing, Qouchan University of Medical Sciences, Qouchan 9471954643, Iran; ghasemiam@mums.ac.ir; 3Addiction and Behavioral Sciences Research Center, North Khorasan University of Medical Sciences, Bojnurd 9414974877, Iran; alireza.afsharisafavi@gmail.com; 4Department of Neurology, School of Medicine, Mashhad University of Medical Sciences, Mashhad 9138813944, Iran; nahayati1360@yahoo.com; 5Student Research Committee, Golestan University of Medical Sciences, Gorgan 4918936316, Iran; 6Student Research Committee, Tehran University of Medical Sciences, Tehran 9691243225, Iran; yavarie@gmail.com; 7Clinical Biochemistry Research Center, Basic Health Sciences Institute, Shahrekord University of Medical Sciences, Shahrekord 8815713471, Iran; mahanfarzan.iri@gmail.com (M.F.); mahour.f9731@gmail.com (M.F.); 8Department of Neurology, Shahrekord University of Medical Sciences and Health Services, Shahrekord 8815713471, Iran; dr_bayati90@yahoo.com; 9The George Institute for Global Health, University of New South Wales, Sydney 90085953331, Australia; faraidoonhaghdoost@gmail.com

**Keywords:** migraine, validity attributes, ID-migraine questionnaire, Persian version

## Abstract

**Background:** Migraine is often not diagnosed or treated properly, despite being a common condition. The ID-migraine is a brief, self-administered test developed as a valuable tool for screening and diagnosing migraine in primary care settings. The objective of the present study was to produce a Persian translation of the original ID-migraine, perform cultural adaptation, and evaluate its validity. **Methods:** Consecutive patients who attended two neurology clinics for headache were enrolled in the study. Diagnoses were established by headache specialists and were compared with the Persian ID-migraine results. **Results:** Among the 657 participants included, 470 (71.5%) were clinically diagnosed with migraine, 120 (18.2%) with tension-type headache, and 40 (6.1%) with cluster headache. The validity attributes of the Persian ID-migraine were as follows: sensitivity, 0.96 (95% CI, 0.93–0.98); specificity, 0.46 (95% CI, 0.35–0.56); positive predictive value, 0.86 (95% CI, 0.82–0.89); negative predictive value, 0.76 (95% CI, 0.63–0.87), and a misclassification error of 14.9%. The questionnaire’s Kappa coefficient was 0.78. **Conclusions:** The Persian version of the ID-migraine questionnaire exhibited sufficient sensitivity and positive predictive value, along with an acceptable misclassification error. However, it demonstrated a deficient level of specificity and a considerably reduced negative predictive value.

## 1. Background

Migraine has a high prevalence, affecting 14.4% of the population worldwide [1]. The proportion of adults affected by migraine in Iran varies from 5.4% to 41.6%, with a pooled prevalence of approximately 15% [2,3]. Migraine was classified according to ICHD criteria as episodic (<15 headache days per month) or chronic (≥15 headache days per month for >3 months), in line with established definitions [4]. Common migraine symptoms include moderate-to-severe, often pulsating headache (frequently one-sided), photophobia, phonophobia, osmophobia, nausea and vomiting, visual disturbances or aura (flashing lights or fortification spectra), and an urge to lie down or avoid movement [5,6]. Migraine predominantly affects young, employed, and productive individuals [7]. According to research on the Global Burden of Disease, migraine was the second leading cause of disability worldwide by 2019 [8]. Despite the significant adverse effects on individuals, approximately 50% of people with migraine are formally diagnosed with this condition by healthcare professionals [9,10]. Several factors contribute to the underdiagnosis and undertreatment of migraine. A significant issue is that many individuals experiencing severe headaches do not seek medical advice, preventing them from receiving a migraine diagnosis [9]. Additionally, the limited duration of consultations and a lack of sufficient knowledge among primary care physicians about the diagnostic criteria established by the International Headache Society (IHS) represent significant challenges to accurate diagnosis and management [11,12,13].

Differentiating migraine from other primary headache conditions is a clinical and public health priority, as a correct diagnosis enables the selection of appropriate acute and preventative therapies while avoiding medicines that are ineffective or potentially harmful for a specific headache type [14]. Precise classification also reduces risks, such as improper long-term analgesic usage and medication-overuse headache, explains prognosis, and identifies individuals who need specialist referral or additional diagnostic testing [15]. For these reasons, authoritative clinical recommendations underline the importance of recognising this difference in daily practice [16]. To aid in the identification and recognition of migraine within primary care settings, Lipton et al. (2003) devised a concise and efficient self-administered identification of migraine (ID-migraine) questionnaire [17]. Three screening questions about the last three months were included in the survey, focusing on key domains such as headache-related disability, nausea, and sensitivity to light. A positive (yes) response to at least two of these questions suggests a likely migraine diagnosis [17].

The ID-migraine questionnaire has been effectively used across specialties, not only in primary care but also in neurology clinics, headache centres, emergency wards, and clinics specialising in temporomandibular and orofacial pain, as well as ophthalmic and otolaryngological facilities. It has been proven valid and reliable for both adults and adolescents, and it has been used in migraine epidemiology and genetic research [12,18,19,20,21,22,23,24,25,26]. The questionnaire has been validated in several countries, including Italy [19], Portugal [21], Turkey [27], Hungary [28], and Latin America [29], where the results were acceptable (Table 1). Additionally, it is important to note that this test cannot solely diagnose migraine, and a comprehensive neurologic examination is also necessary. The validity aspects of the Persian version of the ID-migraine questionnaire are demonstrated in this study, including among participants for whom Persian is not their first language. This target was motivated by the potential linguistic and cultural effects on symptom reporting and screener performance.

## 2. Methods

### 2.1. Patients and Study Design

New patients with a complaint of headache (no history of migraine diagnosis) referred to the Neurology Clinic of Mashhad University of Medical Sciences (Eastern part of Iran) and Imam Ali Hospital, Shahrekord (Western part of Iran), were recruited. Both centres followed the same methodology and protocol.

### 2.2. Inclusion and Exclusion Criteria

Study participants were eligible if they were aged 18 to 65 years, had no history of migraine diagnosis, were able to read and write in Persian, were willing to participate in the study, experienced two or more headache attacks in the past 3 months, and were visiting the medical centre for the first time seeking consultation for headache. The exclusion criteria included unwillingness to continue participation, being under 18 years, having uncontrolled medical or psychiatric conditions, illiteracy, having headache syndromes with no clear diagnosis or not fulfilling the International Classification of Headache Disorders, version three (ICHD-3) diagnostic criteria, pre-existing neurological conditions that could cause secondary headaches, significant cognitive impairment, inability to complete a self-administered questionnaire, and having more than one headache type to ensure responses related to a single headache category.

Exclusion criteria (including headache syndromes without a clear diagnosis) were applied after the initial screening questionnaire and during the diagnostic interview performed by a neurologist using the ICHD-3 criteria.

### 2.3. Sample Size

For studies validating a scale, Comery and Lee [30] suggested that data quality can be categorised by sample size as follows: “100 = poor, 200 = fair, 300 = good, 500 = very good, ≥1000 = excellent.” Anthoine et al. [31] conducted a review of publications on newly developed patient-reported outcome measures, finding that the subject-to-item ratio varies significantly, ranging from a minimum of one to a maximum of 527. Consequently, to ensure a sufficient sample size, we recruited all patients at the two centres over a one-year period who provided informed consent, achieving a target of at least 600 participants for our study.

All patients were asked to complete the questionnaire before their medical visit. They also completed a form regarding their sociodemographic status and clinical characteristics of their headaches. Neurologists’ clinical diagnoses based on the ICHD3 served as the gold standard [6]. In line with the original English version [17], those who responded “yes” to at least two of the three screening questions on the Persian version of the ID-migraine were diagnosed with migraine.

### 2.4. Developing the Persian Version of ID-Migraine

First, the original version of the ID-migraine was translated into Persian by A.I., a bilingual person with excellent proficiency in both Persian and English. The Persian version of ID-migraine was distributed among 15 patients suffering from headache and 10 healthcare workers to assess whether they understood the meaning of each item. The Persian version was then translated into English by Dr. H.B., a linguist. Finally, the English version of the ID-migraine was compared with the original version to check whether it conveyed the same meaning.

### 2.5. Assessment

The ID-migraine questionnaire responses were evaluated in comparison with the clinical diagnosis of migraine. In cases where an individual was diagnosed with migraine, regardless of the presence of other headache disorders, they were considered migraine patients. The doctor’s diagnosis, based on ICHD-3 criteria, served as the reference standard for calculating the sensitivity, specificity, positive predictive value (PPV), negative predictive value (NPV), and misclassification error of the questionnaire. The misclassification error was defined as the proportion of incorrect classifications made by the diagnostic instrument (e.g., if 100 subjects are classified and 20 are misclassified, the misclassification error = 20/100 = 0.20, or 20%). These performance metrics were calculated for each item in the ID-migraine questionnaire. Subgroup analyses were performed according to participants’ mother tongue (Persian, Kurdish, Arabic), sex (male, female), age (≤30, 31–50, >50 years), and disease duration (≤48 months, >48 months). The Persian version of the ID-migraine questionnaire is provided in the Appendix A.

### 2.6. Test–Retest Reliability

Eighty-five participants completed the ID-migraine questionnaire twice to assess its test–retest reliability. The interval between the first and second completions of the ID-migraine Questionnaire was set at three months. The assessment was conducted using the widely accepted Cohen’s Kappa metric. According to the literature, the level of agreement for values of Cohen’s Kappa was considered as follows [32]: 0.00 indicates poor agreement, 0.00–0.20 indicates slight agreement, 0.21–0.40 indicates fair agreement, 0.41–0.60 indicates moderate agreement, 0.61–0.80 indicates substantial agreement, and 0.81–1.00 indicates almost perfect agreement.

### 2.7. Data Analysis

Data were analysed using SPSS Statistics Version 18 (IBM SPSS Statistics for Windows, Version 18. Armonk, NY, USA: IBM). The Kappa coefficient of the questionnaire was calculated using SPSS. MedCalc software was used to determine the validity characteristics of the ID Migraine (sensitivity, specificity, PPV, and NPV), confidence intervals, and misclassification error [33].

### 2.8. Ethics

The Shahrekord University of Medical Sciences’ ethics committee approved the study protocol (IR.SKUMS.REC.1401.137).

## 3. Results

A total of 657 Iranian patients with headache complaints were enrolled in this study (Table 2). The mean age (SD) was 48.8 (17.0) years, and the majority were females (60.7%).

Based on the neurologist’s diagnosis, the distribution of headache disorders was reported as follows: episodic migraine (*n* = 377; 57.4%), chronic migraine (*n* = 93; 14.2%), episodic tension-type headache (*n* = 58; 8.8%), chronic tension-type headache (*n* = 62; 9.4%), cluster headache (*n* = 40; 6.1%), and other types (*n* = 27; 4.1%) in the total sample (N = 657). “Other” included conditions such as headache associated with sexual activity and medication-overuse headache. The diagnostic performance of the ID-migraine screening tool compared with the gold-standard diagnosis (ICHD-3–based neurologist assessment) is presented in Table 3.

The quality scores of the Persian ID-migraine questionnaire, categorised by the mother tongue, are summarised in Table 4. The original cutoff value was defined as more than two “yes” responses out of the three screening questions. Considering all participants, the sensitivity, specificity, positive predictive value (PPV), and negative predictive value (NPV) were 0.84, 0.39, 0.78, and 0.50, respectively. In addition, the misclassification rate was estimated to be 28.4%. The participants whose mother tongue was Persian had higher diagnostic criteria as follows: sensitivity 0.96 (95%CI: 0.93–0.98), specificity 0.46 (95%CI: 0.35–0.56), PPV 0.86 (95%CI: 0.82–0.89), and NPV 0.76 (95%CI: 0.63–0.87).

Photophobia, disability, and nausea demonstrated high sensitivity and PPV (>0.75). Among all participants, the screening question on disability exhibited the highest sensitivity (0.92), specificity (0.52), and negative predictive value (0.72), whereas the question on photophobia demonstrated the highest positive predictive value (0.87).

Among participants with Persian as their mother tongue, the screening question on disability exhibited the highest sensitivity (0.96), positive predictive value (0.87), and negative predictive value (0.77), while the question on photophobia demonstrated the highest specificity (0.52). (Table 5).

The clinical features of headaches in non-migraine individuals are displayed in Table 6. Of those with tension-type headache, 69 (57.5%) had a positive ID-migraine test result (31 with chronic and 38 with episodic tension-type headache). Among these participants, the most frequent complications were nausea (95.6%) and photophobia (82.6%). In the group with cluster headache, 29 (72.5%) patients (all episodic) had positive ID-migraine results. Nausea (93.1%) and photophobia (93.1%) were the most common symptoms in this group.

Table 7 shows the questionnaire quality ratings for clinically relevant subgroups. There was no remarkable difference between men and women. While the sensitivity, specificity, and PPV of the questionnaire remained consistent among patients with disease durations of less than 48 months compared to those with durations exceeding 48 months, significant differences were observed in NPV.

## 4. Discussion

This study aimed to validate and cross-culturally adapt the Persian version of the ID-migraine questionnaire. With a sensitivity of 0.84, specificity of 0.39, and a kappa coefficient of 0.78 for test–retest reliability assessment, the Persian adaptation of the ID-migraine Questionnaire demonstrated validity and reliability as a screening instrument for migraine. Nonetheless, preliminary appraisal indicates that its quality metrics are inferior to those of the original instrument and other validated versions [17,19,21,27,28,29,34,35,36,37]. However, after a subgroup analysis based on the mother tongue of participants, the results for those whose mother tongue was Persian were similar to those observed in previous research and were consistent with the reliability of the original version of ID-migraine by Lipton et al. [17]. This highlights the necessity of translating diagnostic instruments to accommodate the local languages of people, even in countries with an official language, particularly in nations with cultural and linguistic diversity. Iran is a linguistically diverse country with Persian (Farsi) as its official language. However, many individuals speak regional languages—such as Kurdish, Azeri, Gilaki, Arabic, and others—as their first language. Although Persian is widely spoken, linguistic diversity can influence how well individuals understand and respond to screening tools such as the ID-migraine. Cultural and linguistic adaptation for multilingual populations poses additional challenges, as it requires accounting for dialectal variations and different cultural understandings of health-related questions [38]. We acknowledge that future adaptations of the ID-migraine could benefit from tailoring it to the various languages spoken in Iran to enhance its accuracy and reduce the misclassification rate among non-Persian speakers.

In our study, the high sensitivity of the Persian version of the ID-migraine (0.96) indicates that this questionnaire accurately identifies a large proportion of individuals with migraine. In other words, it has a low rate of false negatives, ensuring that most individuals with migraine are accurately identified. Its moderate specificity (0.46) indicates the ability of the questionnaire to identify those without migraine. The moderate specificity suggests a higher rate of false positives, meaning that some individuals without migraine might be incorrectly classified as having migraine. We suppose that potential reasons for the moderate specificity could include the overlap of symptoms with other headache disorders and the cultural factors. Furthermore, somatisation, which is the expression of psychological distress through physical symptoms, may be more prevalent in some cultures [39,40]. This could influence the responses to the questionnaire and result in false positives.

Arabic-speaking patients showed the lowest sensitivity and specificity on the ID-migraine questionnaire compared with Persian speakers. This may result from cultural differences in pain expression and reporting (stoicism vs. somatisation), as well as translation and semantic issues for key terms (e.g., “nauseated”, “disability”), which can cause misinterpretation. The findings indicate that the questionnaire needs cultural adaptation beyond literal translation to improve its diagnostic accuracy in Arabic-speaking populations [41]. The observed differences in sensitivity and specificity highlight the importance of considering cultural diversity, linguistic nuances, and traditional beliefs when developing and validating screening tools for conditions such as migraine. Future research should focus on the culturally sensitive adaptation of the questionnaire, involving bilingual experts and community stakeholders, to ensure its validity and applicability across diverse populations.

Accordingly, we suggest that the questionnaire be used as a screening tool alongside a comprehensive clinical evaluation, including patient history, physical examination, and potentially other diagnostic tests. By understanding these strengths and limitations, healthcare professionals can effectively utilise the ID-migraine questionnaire as part of a comprehensive diagnostic approach for migraine in Persian-speaking populations.

Compared with previous validation studies, the Persian ID-migraine questionnaire showed similar sensitivity to the Turkish [27], Italian [19], Portuguese [21], Hungarian [28], German [35], and Spanish [29] versions and higher sensitivity than the English [17], French [36], and Chinese [34] versions (Table 1). Regarding specificity, our results were similar to those of the Hungarian [28], Turkish [27], and Chinese [34] studies but lower than those of the others (Table 1).

In our study and other versions of the ID-migraine, such as the Turkish translation [21], migraine was misclassified as other headache disorders, particularly tension-type headache. This misclassification could be due to the fact that some participants might have had both disorders, as it is possible for different headache disorders, such as migraine and tension-type headache, to coexist. Additionally, these two conditions share several overlapping symptoms, which can make it challenging for participants to accurately distinguish between them. The ID-migraine questions may not always be sufficient to differentiate between these disorders, especially in chronic cases. Moreover, cultural or linguistic factors may further impact the interpretation of the questions, contributing to misclassification in different populations, such as Persian or Turkish speakers. This highlights the need for further refinements to improve the specificity of the tool and to better accommodate cultural and linguistic diversity.

In a study conducted by Sahu et al. [37], the ID-migraine was translated into two languages to accommodate participants who spoke different languages despite being from the same country. Their participants were composed of two North Indian vernacular languages, Hindi and Punjabi. The sensitivity of the Hindi version of ID-migraine was 94% (95% CI, 79% to 99%); specificity, 56% (95% CI, 31% to 78%). The Punjabi version demonstrated a sensitivity of 86% (95% CI, 68% to 96%); specificity, 43% (95% CI, 23% to 66%). The specificity of the Punjabi version was found to be low in comparison to the Persian and Hindi versions. This could be due to differences between the two populations. Sahu et al. believed that the relatively lower specificities in the face of high sensitivities are perfectly acceptable for a screening tool, as opposed to a diagnostic instrument, which should have both high sensitivity and specificity [37].

Future research on the ID-migraine screening tool could explore several important areas to further enhance its utility and effectiveness in clinical practice. Studies aimed at identifying the factors contributing to false positive results on the ID-migraine questionnaire would provide valuable insights. By understanding the reasons behind these false positives, such as comorbidities or overlapping symptoms with other conditions, researchers could refine the tool to improve its specificity, thereby reducing the rate of misdiagnosis. Furthermore, evaluating how well the ID-migraine screening tool works in different healthcare settings, such as primary care, specialty clinics, and emergency departments, could offer a more thorough insight into its practical usefulness in the real world. One Italian ED study demonstrated that, after excluding secondary headaches, ID-migraine maintained excellent accuracy in patients with headache. Among ED outpatients, Mostardini et al. observed 94% sensitivity, 83% specificity, and 99% PPV [22]. In primary care, Cousins et al. found a pooled sensitivity of 0.84 and specificity of 0.76 [20]. In other words, an ID-migraine negative result markedly lowers the likelihood of migraine, making it useful for screening non-migraine headaches. Our study on Persian ID-migraine similarly found robust sensitivity and predictive values, which aligned with these primary care results.

### 4.1. Implications for Clinical Practice

Validating the Persian version of the ID-migraine Questionnaire has significant implications for healthcare practices in Iran and other multilingual populations. The results of this study underscore the critical importance of providing culturally and linguistically accurate diagnostic tools to ensure the precise evaluation and effective treatment of migraine.

In countries with diverse language environments, it is essential to offer diagnostic tools in multiple languages to reach a broader range of individuals and avoid excluding patients because of language obstacles. This inclusive approach helps ensure that more people have access to accurate migraine diagnosis, which is the first step towards effective management and treatment.

### 4.2. Limitations and Strengths

A major strength of the present validation study lies in its substantial sample size, which minimises the influence of individual variability and potential outliers on the overall data. Another strength of this study is the recruitment of patients from two clinics. This approach helps improve the representativeness of the sample by including patients from different settings, potentially with varying demographic characteristics and clinical presentations. This diversity in the sample enhances the external validity of the study, making the results more applicable to a wider population of patients with migraine.

A principal limitation of the present study is that participants were recruited exclusively from specialist headache services; therefore, the sample may not be representative of the general population. Consequently, regardless of diagnostic classification, individuals with more severe head pain and accompanying symptoms are likely to be overrepresented. In addition, we did not consider the presence or absence of aura with migraine in our analysis, which could provide additional insights. Furthermore, we suggest that researchers obtain data on probable migraine diagnoses in their studies.

## 5. Conclusions

These findings enable us to have a screening tool for diagnosing migraine that has been validated and adapted for cross-cultural use in Persian-speaking populations. It is hoped that the Persian version of the ID-migraine will enhance early diagnosis and lessen the impact on patients’ quality of life. We found that the Persian ID-migraine tool functions differently in Iranians whose mother tongue is not Persian. We suggest that researchers conduct further validation studies in other languages prevalent in multilingual societies to ensure the reliability and validity of the questionnaire across diverse populations. Recognising and addressing the impact of different mother tongues on the performance of diagnostic tools is crucial. This approach ensures that all patients, regardless of their linguistic background, have access to accurate and reliable diagnostic resources, ultimately improving healthcare outcomes and equity in healthcare access.

## Figures and Tables

**Table 1 medsci-13-00213-t001:** Results of the ID-migraine questionnaire’s validation in both the original and other languages.

Language	Author	Country	Year	Number of Patients	Sensitivity (95% CI)	Specificity (95% CI)	Positive Predictive Value (95% CI)	Negative Predictive Value (95% CI)
English	Lipton et al.	USA	2003	563	81% (77% to 85%)	75% (64% to 84%)	93% (89.9% to 95.8%)	(Nil)
Turkish	Karli et al.	Turkey	2007	3682	91.8% (Nil)	63.4% (Nil)	71.9% (Nil)	88.4% (Nil)
Italian	Brighina et al.	Italy	2007	220	95% (91% to 98%)	72% (62% to 82%)	88% (82% to 93%)	87% (78% to 95%)
Portuguese	Gil- Gouveia R & Martins	Portugal	2010	142	94% (87% to 97%)	60% (46% to 73%)	80% (71% to 87%)	85% (70% to 94%)
French *	Streel et al.	France	2015	751	87.5% (Nil)	100% (Nil)	100% (Nil)	93.5% (Nil)
Chinese	Wang et al.	China	2015	555	84.0 (75.0% to 90.0%)	64.0% (59.0% to 68.0%)	Nil	Nil
Hungarian	Csépány et al.	Hungary	2018	380	95% (92% to 97%)	42% (31% to 55%)	88% (84% to 91%)	65% (50% to 78%)
German *	Thiele et al.	Germany	2020	105	99% (Nil)	68% (Nil)	90% (Nil)	95% (Nil)
Spanish	Rodriguez-Rivas R & Martinez CM.	Spain	2022	115	91.7% (74.2% to 97.7%)	82.5% (68.1% to 91%)	75.9% (57.9% to 87.8%)	Nil
Hindi	Sahu et al.	India	2023	100	94% (79% to 99%)	56% (31% to 78%)	79% (69% to 86%)	83% (55% to 95%)
Punjabi	Sahu et al.	India	2023	100	86% (68% to 96%)	43% (23% to 66%)	68% (58% to 76%)	69% (44% to 86%)

* Extended versions of ID-migraine in French and German are available for translation and validation.

**Table 2 medsci-13-00213-t002:** Demographic and clinical characteristics of the participants.

Age (year); mean (SD)		48.8 (17.0)
Duration of symptoms (month); median (Q_1_, Q_3_)		43 (24, 63)
Sex; n (%)	Male	258 (39.3)
	Female	399 (60.7)
Mother tongue ^Ω^; n (%)	Persian	423 (64.4)
	Arabic	73 (11.1)
	Kurdish	142 (21.6)

^Ω^: Of 657 participants, 19 (2.9%) patients were originally from Afghanistan but were not included in the subgroup analysis due to the low number.

**Table 3 medsci-13-00213-t003:** 2 × 2 contingency table, ID-migraine vs. gold-standard diagnosis (ICHD-3).

	Gold-Standard *: Migraine	Gold-Standard: No migraine	Total
ID-Migraine positive	TP ^a^ = 396	FP ^b^ = 113	509
ID-Migraine negative	FN ^c^ = 74	TN ^d^ = 74	148
Total	470	187	657

*: Doctor diagnosis was established according to the International Classification of Headache Disorders, 3rd edition (ICHD-3) criteria; ^a^. True Positive; ^b^. False Positive; ^c^. False Negative; ^d^. True Negative.

**Table 4 medsci-13-00213-t004:** Quality scores of the Persian ID-migraine questionnaire categorised by the participants’ mother tongue.

	Positive ID-Migraine (≥2 “yes”)				
Mother tongue (n)	Misclassification *	NPV (95% CI)	PPV (95% CI)	Specificity (95% CI)	Sensitivity (95% CI)
Persian (423)	14.9%	0.76 (0.63–0.87)	0.86 (0.82–0.89)	0.46 (0.35–0.56)	0.96 (0.93–0.98)
Arabic (73)	69.8%	0.27 (0.11–0.48)	0.31 (0.19–0.47)	0.18 (0.07–0.33)	0.44 (0.27–0.62)
Kurdish (142)	45.7%	0.36 (0.25–0.49)	0.68 (0.57–0.78)	0.48 (0.33–0.62)	0.57 (0.46–0.67)
All Iranians (657)	28.4%	0.50 (0.42–0.58)	0.78 (0.74–0.81)	0.39 (0.32–0.47)	0.84 (0.80–0.87)

PPV: positive predictive value, NPV: negative predictive value, and 95% CI: 95% confidence interval. * Misclassification error is the percentage of predictions a diagnostic instrument is wrong in diagnosing medical conditions. It is calculated by dividing the number of incorrect diagnoses by the total number of diagnoses of the condition.

**Table 5 medsci-13-00213-t005:** Individual quality scores for each participant’s items on the Persian version of the ID-migraine (*n* = 657) and for participants with Persian mother tongue (*n* = 423).

	Sensitivity (95% CI)	Specificity (95% CI)	PPV (95% CI)	NPV (95% CI)	Misclassification
All participants					
Nausea	0.82 (0.78–0.85)	0.41 (0.34–0.48)	0.78 (0.75–0.80)	0.47 (0.41–0.53)	29.9%
Photophobia	0.87 (0.82–0.90)	0.52 (0.41–0.63)	0.87 (0.82–0.90)	0.52 (0.41–0.63)	31.9%
Disability	0.92 (0.89–0.94)	0.52 (0.45–0.60)	0.83 (0.81–0.85)	0.72 (0.65–0.78)	19.3%
ID-migraine positive (≥2 “yes”)	0.84 (0.81–0.87)	0.40 (0.33–0.47)	0.78 (0.76–0.80)	0.50 (0.43–0.57)	28.5%
Persian mother tongue					
Nausea	0.91 (0.87–0.94)	0.46 (0.35–0.56)	0.86 (0.81–0.89)	0.58 (0.46–0.69)	18.9%
Photophobia	0.87 (0.82–0.90)	0.52 (0.41–0.63)	0.87 (0.82–0.90)	0.52 (0.41–0.63)	20.8%
Disability	0.96 (0.93–0.98)	0.48 (0.37–0.58)	0.87 (0.83–0.90)	0.77 (0.64–0.87)	14.4%
ID-migraine positive (≥2 “yes”)	0.96 (0.93–0.98)	0.46 (0.35–0.56)	0.86 (0.82–0.89)	0.76 (0.63–0.87)	14.9%

**Table 6 medsci-13-00213-t006:** The non-migraine subjects’ self-reported headache features.

	ID Migraine	n	Worse with Movement	Nausea	Vomiting	Photophobia	Phonophobia
Tension-type headache	positive	69	41 (59.4%)	66 (95.6%)	15 (21.7%)	57 (82.6%)	31 (44.9%)
	negative	51	40 (78.4%)	0	6 (11.7%)	0	8 (15.6%)
Cluster headache	positive	29	16 (55.1%)	27 (93.1%)	1 (3.4%)	27 (93.1%)	15 (5.7%)
	negative	11	11 (100%)	0	2 (18.1%)	0	4 (36.3%)
Other headache-type	positive	15	8 (53.3%)	14 (93.3%)	2 (13.3%)	14 (93.3%)	5 (33.3%)
	negative	12	5 (41.6%)	0	1 (8.3%)	0	5 (41.6%)

Numbers are n (%).

**Table 7 medsci-13-00213-t007:** Quality scores of the Persian version of the ID-migraine questionnaire in the clinically relevant subgroups for all participants (*n* = 657) and participants with Persian mother tongue (*n* = 423).

		n (%)	Sensitivity (95% CI)	Specificity (95% CI)	PPV (95% CI)	NPV (95% CI)	Misclassification
All participants							
Sex							
	Men	258 (39.3)	0.84 (0.78–0.89)	0.39 (0.28–0.51)	0.78 (0.75–0.81)	0.49 (0.38–0.60)	28.3%
	Women	399 (60.7)	0.84 (0.79–0.88)	0.40 (0.31–0.50)	0.78 (0.75–0.80)	0.51 (0.42–0.59)	28.6%
Age							
	≤30	131 (19.9)	0.83 (0.74–0.90)	0.42 (0.26–0.61)	0.82 (0.77–0.86)	0.43 (0.30–0.58)	26.7%
	30–50	210 (32)	0.86 (0.79–0.91)	0.43 (0.31–0.55)	0.76 (0.72–0.80)	0.59 (0.47–0.70)	28.1%
	>50	316 (48.1)	0.84 (0.78–0.88)	0.36 (0.26–0.47)	0.77 (0.74–0.80)	0.46 (0.37–0.56)	29.4%
Duration of disease							
	≤48	365 (55.6)	0.87 (0.82–0.91)	0.42 (0.33–0.51)	0.74 (0.71–0.77)	0.62 (0.52–0.70)	28.7%
	>48	292 (44.4)	0.82 (0.76–0.87)	0.36 (0.24–0.47)	0.82 (0.79–0.85)	0.34 (0.25–0.45)	28.1%
Participants with Persian mother tongue							
Sex							
	Men	166 (39.2)	0.95 (0.90–0.98)	0.45 (0.28–0.63)	0.88 (0.81–0.92)	0.71 (0.48-.89)	14.4%
	Women	257 (60.8)	0.96 (0.93–0.98)	0.46 (0.33–0.59)	0.86 (0.80–0.90)	0.79 (0.62–0.91)	15.1%
Age							
	≤30	83 (19.6)	0.96 (0.88–0.99)	0.43 (0.18–0.71)	0.89 (0.79–0.95)	0.67 (0.29–0.93)	13.2%
	30–50	137 (32.4)	0.95 (0.90–0.99)	0.48 (0.32–0.66)	0.83 (0.75–0.90)	0.82 (0.59–0.95)	16.8%
	>50	203 (48)	0.96 (0.92–0.99)	0.44 (0.28–0.60)	0.87 (0.81–0.92)	0.75 (0.53–0.90)	14.3%
Duration of disease							
	≤48	247 (58.4)	0.96 (0.93–0.99)	0.48 (0.36–0.60)	0.81 (0.75–0.87)	0.85 (0.71–0.94)	17.8%
	>48	176 (41.6)	0.95 (0.91–0.98)	0.37 (0.16–0.62)	0.92 (0.87–0.96)	0.50 (0.23–0.77)	10.8%

Test–retest reliability assessed using Cohen’s kappa statistic showed substantial agreement between the two administrations of the ID-migraine questionnaire (*n* = 85, K = 0.78; *p* < 0.001).

## Data Availability

The original contributions presented in this study are included in the article/Appendix A. Further inquiries can be directed to the corresponding author.

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
