# Peer review of "The Validation of the Persian Version of the ID-Migraine Questionnaire"

_medsci, 2025, doi:10.3390/medsci13040213_

Round 1

Reviewer 1 Report

Comments and Suggestions for Authors

The manuscript is about creating a diagnostic tool for migraine adapted to their own country (screener). This is extremely important and valuable. It is particularly desirable that they use the ID migraine used in the United States as a reference, but Japan has also created a 4-item migraine screener (Takeshima T, for Study Group for Optimal Headache M. A simple migraine screening instrument the validation study in Japan. Cephalalgia. 2005;25(10):970.). This paper only has the abstract for Cephalalgia, so I think it was difficult to find. One additional item has been added to the ID migraine. This item is related to olfaction, osmophobia. This is because many Japanese migraine patients have hyperolfactory sensitivity. This manuscript should also be cited.

Author Response

Comments 1: It is particularly desirable that they use the ID migraine used in the United States as a reference, but Japan has also created a 4-item migraine screener (Takeshima T, for Study Group for Optimal Headache M. A simple migraine screening instrument the validation study in Japan. Cephalalgia. 2005;25(10):970.). This paper only has the abstract for Cephalalgia, so I think it was difficult to find. One additional item has been added to the ID migraine. This item is related to olfaction, osmophobia. This is because many Japanese migraine patients have hyperolfactory sensitivity.

Response: Thank you for pointing us to Takeshima et al. (Cephalalgia, 2005). We attempted to obtain the full text through multiple library and publisher channels, but only the abstract is publicly available and we were unable to access the complete article. Because the full paper could not be reviewed, we could not verify the study’s methods and items in sufficient detail to cite it as evidence in the Introduction.

As an alternative, we reviewed other publications by the same author and identified a recent 2023 study (Takeshima co-author) that presents a validated Japanese screening questionnaire; that study’s questionnaire (new version) does not include an item on osmophobia. If the reviewer can provide a copy of the 2005 full text (or a DOI/link), we will immediately review it and, if appropriate, add and discuss it in the Introduction.

Reference: Tanobe K, Machida M, Motoya R, Takeoka A, Danno D, Miyahara J, Takeshima T, Kumano H, Tayama J. Development and Validation of a Japanese-Language Questionnaire to Screen for Tension-Type Headaches and Migraines. Cureus. 2023 Sep 4;15(9):e44633. doi: 10.7759/cureus.44633.

Reviewer 2 Report

Comments and Suggestions for Authors

This is a useful piece of research to test and validate the ID-migraine screener for the Persian language.

The Background needs to introduce one of the main aims of the paper (deduced by the results and discussion) which is to test the ID-migraine test in Persian with those who do not have Persian as a mother tongue/first language. Also need to introduce/define episodic/chronic migraine and discuss the usefulness/need for the ID-migraine to distinguish between migraine another primary headache disorders.

The Methods/Results sections need some additional detail and clarification. At what point was the exclusion criteria of 'headache syndromes with no clear diagnosis or not fulfilling ICHD criteria' applied? If medication overuse headache is an exclusion, why is this category included in Table 3? In section 2.5, the use of the 2nd person pronoun is unusual and should be revised. Please provide the ABCD table with the numbers of people with gold-standard diagnosed migraine against those with positive/negative ID-migraine test.  The numbers in the text (lines 163-164) do not appear to match with the numbers in Table 3 or the estimates for specificity, sensitivity etc, e.g. if 308 of those with positive ID-migraine have migraine and there are 470 with migraine that would give a sensitivity of 66% not 84%.

Please explain why Table 3 categorises people with episodic migraine and chronic migraine when this is not a feature of the ID-migraine. 

The discussion is extremely repetitive and should be revised and condensed. There are multiple paragraphs with no references which is inappropriate. If recommending how the ID-migraine works in different settings (primary care, ED, etc) studies that have done this should be referenced and discussed. 

Author Response

Thank you for these constructive and precise comments. We addressed each point as follows:

Comments 1: The Background needs to introduce one of the main aims of the paper (deduced by the results and discussion) which is to test the ID-migraine test in Persian with those who do not have Persian as a mother tongue/first language:

Response: We have revised the Introduction to explicitly state that one of the primary aims is to validate the ID-Migraine in Persian, including among participants whose first language is not Persian (line 84-88).

“In the present study we exhibit the validity features of the Persian version of the ID-migraine questionnaire, including among participants for whom Persian is not the first language. This target was motivated by potential linguistic and cultural effects on symptom reporting and screener performance”

Comments 2: need to introduce/define episodic/chronic migraine:

Response: we added a brief definition of episodic vs chronic migraine per ICHD (line 49-51).

“Migraine was classified according to ICHD criteria as episodic (<15 headache days per month) or chronic (≥15 headache days per month for >3 months), in line with established definitions”

reference: Lipton RB, Silberstein SD. Episodic and chronic migraine headache: breaking down barriers to optimal treatment and prevention. Headache. 2015;55 Suppl 2:103-22; quiz 23-6. 10.1111/head.12505_2.

Comments 3: discuss the usefulness/need for the ID-migraine to distinguish between migraine another primary headache disorders:

Response: The correction was made on lines 62-69.

“Differentiating migraine from other primary headache conditions is a clinical and public-health priority, as a correct diagnosis allows for the selection of suitable acute and preventative therapy while avoiding the use of medicines that are ineffective or dangerous for a specific headache type (12). Precise classification also reduces risks such as improper long-term analgesic usage and medication-overuse headache, ex-plains prognosis, and identifies individuals who need specialist referral or additional diagnostic testing (13). For these reasons, authoritative clinical recommendations underline the importance of recognizing this difference in everyday practice (14).”

reference:

  1. Diener H-C, Holle-Lee D, Nägel S, Dresler T, Gaul C, Göbel H, et al. Treatment of migraine attacks and prevention of migraine: Guidelines by the German Migraine and Headache Society and the German Society of Neurology. Clinical and Translational Neuroscience. 2019;3(1):2514183X18823377.
  2. Lipton RB, Seng EK, Chu MK, Reed ML, Fanning KM, Adams AM, et al. The Effect of Psychiatric Comorbidities on Headache-Related Disability in Migraine: Results from the Chronic Migraine Epidemiology and Outcomes (CaMEO) Study. Headache. 2020;60(8):1683-96. 10.1111/head.13914.
  3. Robbins MS. Diagnosis and management of headache: a review. Jama. 2021;325(18):1874-85.

Comments 4: At what point was the exclusion criteria of 'headache syndromes with no clear diagnosis or not fulfilling ICHD criteria' applied?

Response: Exclusion criteria (including headache syndromes without a clear diagnosis) were applied after the initial screening questionnaire and during the diagnostic interview performed by a neurologist using ICHD-3 criteria. The correction was made on lines 109-111. 

Comments 5: If medication overuse headache is an exclusion, why is this category included in Table 3?

Response: The research team, in order to examine different headache presentations, decided during data analysis not to exclude medication-overuse headache (MOH) and to include it in the results and analyses. However, while drafting the Methods section of the manuscript the team inadvertently followed the original proposal, resulting in a writing error. The correction was made on line 107.

Comments 6: Please provide the ABCD table with the numbers of people with gold-standard diagnosed migraine against those with positive/negative IDmigraine test. The numbers in the text (lines 163-164) do not appear to match with the numbers in Table 3 or the estimates for specificity, sensitivity etc, e.g. if 308 of those with positive IDmigraine have migraine and there are 470 with migraine that would give a sensitivity of 66% not 84%.

Response: The findings presented in Table 3 described in the text. Then, Table 3 was renamed to Table ABCD with the numbers of people with gold-standard diagnosed migraine against those with positive/negative ID-migraine test. The errors have been corrected (lines 171-178)

"based on the neurologist’s diagnosis, the distribution of headache disorders was reported as follows: episodic migraine (n = 377; 57.4%), chronic migraine (n = 93; 14.2%), episodic tension-type headache (n = 58; 8.8%), chronic tension-type headache (n = 62; 9.4%), cluster headache (n = 40; 6.1%), and other types (n = 27; 4.1%) — for the total sample, N = 657. “Other” included conditions such as medication-overuse headache and headache associated with sexual activity. The diagnostic performance of the ID-Migraine screening tool compared with the gold-standard diagnosis (ICHD-3–based neurologist assessment) is presented in Table 3."

Comments 7: In section 2.5, the use of the 2nd person pronoun is unusual and should be revised.

Response: Corrections were made to lines 141–148.

Comments 8: Please explain why Table 3 categorises people with episodic migraine and chronic migraine when this is not a feature of the IDmigraine.

Response: the ID-Migraine screening tool does not distinguish episodic from chronic migraine, we categorized participants by the neurologist’s ICHD-3 diagnosis to document the clinical heterogeneity of our sample. Reporting these clinician-determined categories demonstrates that both episodic and chronic migraine cases were included, allowing the reader to appreciate the broad spectrum of headache presentations assessed in this study.

Comments 9: The discussion is extremely repetitive and should be revised and condensed. There are multiple paragraphs with no references which is inappropriate. If recommending how the ID-migraine works in different settings (primary care, ED, etc) studies that have done this should be referenced and discussed.

Response: We condensed the Discussion, removed repetitive paragraphs and added references.

Round 2

Reviewer 1 Report

Comments and Suggestions for Authors

The reference identified by the reviewers is as follows (attached file):
Takao Takeshima for Study Group for Optimal Headache M. A simple migraine screening instrument.the validation study in Japan. 
Cephalalgia. 2005;25(10):970.

Author Response

Thank you for providing the abstract. Upon closer inspection we found that the questionnaire reported by TAKESHIMA Takao contains 12 items and differs from the instrument used in our study. The abstract appears to have been prepared for a 2005 conference and the full article seems to have been published in 2015; however, we were unable to access the full text. Nevertheless, in the revised manuscript we have cited TAKESHIMA Takao in the Introduction.
"Common migraine symptoms include moderate-to-severe, often pulsating headache (frequently one-sided), photophobia, phonophobia, osmophobia, nausea and vomiting, visual disturbances or aura (flashing lights or fortification spectra), and an urge to lie down or avoid movement"

Reviewer 2 Report

Comments and Suggestions for Authors

Paragraph from line 235 need to specify this is discussing the Persian version of the ID-Migraine.

Author Response

comment: Paragraph from line 235 need to specify this is discussing the Persian version of the ID-Migraine.

Response: The correction was made

"This study aimed to validate and cross-culturally adapt the Persian version of the ID-migraine questionnaire. With a sensitivity of 0.84, specificity of 0.39, and a kappa coefficient of 0.78 for test-retest reliability assessment, the Persian adaptation of the ID-migraine Questionnaire demonstrated validity and reliability as a screening instrument for migraine."